



# Development and intercity transferability of land-use regression models for predicting ambient PM₁₀, PM₂.₅, NO₂ and O₃ concentrations in northern Taiwan

Zhiyuan Li[1], Kin-Fai Ho[2,1], Steve Hung Lam Yim[3,4,1,*]

[1] Institute of Environment, Energy and Sustainability, The Chinese University of Hong Kong, Shatin, N.T., Hong Kong, China

[2] The Jockey Club School of Public Health and Primary Care, The Chinese University of Hong Kong, Shatin, N.T., Hong Kong, China

[3] Department of Geography and Resource Management, The Chinese University of Hong Kong, Shatin, N.T., Hong Kong, China

[4] Stanley Ho Big Data Decision Analytics Research Centre, The Chinese University of Hong Kong, Shatin, N.T., Hong Kong, China

*Correspondence to*: Steve Hung Lam Yim (steveyim@cuhk.edu.hk)

**Abstract.** To provide long-term air pollutant exposure estimates for epidemiological studies, it is essential to test the feasibility of developing land-use regression (LUR) models using only routine air quality measurement data and to evaluate the transferability of LUR models between nearby cities. In this study, we develop and evaluate the intercity transferability of annual average LUR models for ambient respirable suspended particulates (PM₁₀), fine suspended particulates (PM₂.₅), nitrogen dioxide (NO₂), and ozone (O₃) in the Taipei–Keelung metropolitan area of northern Taiwan in 2019. Ambient PM₁₀, PM₂.₅, NO₂, and O₃ measurements at 30 fixed-site stations were used as the dependent variables, and a total of 156 potential predictor variables in six categories (i.e., population density, road network, land-use type, normalized difference vegetation index, meteorology, and elevation) were extracted using buffer spatial analysis. The LUR models were developed using the supervised forward linear regression approach. The LUR models for ambient PM₁₀, PM₂.₅, NO₂, and O₃ achieved relatively high prediction performance, with $R^2$ and leave-one-out cross-validation (LOOCV) $R^2$ values of $> 0.72$ and $> 0.53$, respectively. The intercity transferability of LUR models varied among the air pollutants, with transfer-predictive $R^2$ values of $> 0.62$ for NO₂ and $< 0.56$ for the other three pollutants. The LUR-model-based 500 m $\times$ 500 m spatial distribution maps of these air pollutants illustrated pollution hotspots and the heterogeneity of population exposure, which provide valuable information for policymakers in designing effective air pollution control strategies. The LUR-model-based air pollution exposure estimates captured the spatial variability of exposure for participants in a cohort study. This study highlights that LUR models can be reasonably established upon a routine monitoring network but there exist uncertainties when transferring LUR models between nearby cities. To the best of our knowledge, our study is the first to evaluate the intercity transferability of LUR models in Asia.



# 1 Introduction

Air pollution has been reported to be positively associated with a variety of health effect endpoints, such as lung function and respiratory-disease-related hospital admission (Çapraz et al., 2017; Zhou et al., 2020). Exposure assessment of air pollution is a critical component of epidemiological studies (Cai et al., 2020; Hoek et al., 2008). Cohort studies focusing on the long-term effect on specific diseases of exposure to air pollution require accurate exposure estimates for a large group of participants (e.g., thousands or more) over a defined time period (Brokamp et al., 2019; Morley and Gulliver, 2018; Zhou et al., 2020). Different air quality prediction methods, such as air dispersion models, atmospheric chemical transport models, satellite remote sensing, and various statistical methods, have been developed and applied to estimating population exposure to air pollution (Hao et al., 2016; Michanowicz et al., 2016). Among these exposure assessment methods, land-use regression (LUR) is a standard modeling approach widely used to characterize long-term average air pollutant concentrations at a fine spatial scale, which provides high spatial resolution estimates of exposure for use in epidemiological studies (Bertazzon et al., 2015; Eeftens et al., 2016; Jones et al., 2020).

The LUR method is based on the principle that ambient air pollutant concentrations at fixed-site measurement stations are linearly associated with different environmental features (e.g., land use, population density, road network, and meteorological conditions) surrounding these stations (Anand and Monks, 2017; Lu et al., 2020; Naughton et al., 2018; Wu et al., 2017). In a city or even at a smaller spatial scale area, the LUR method is comparable to or sometimes even better than the approaches of satellite-remote-sensing-based air quality retrievals and air dispersion models in characterizing spatiotemporal variation in air pollution (Marshall et al., 2008; Shi et al., 2020). Following feasible procedures of data processing and analysis, established air pollution LUR models can be applied to predict concentrations of air pollutants at locations without measurements at multiple spatial scales or at residential locations of participants in epidemiological studies (Liu et al., 2016; Shi et al., 2020).

In recent years, a large number of air pollution LUR studies have been conducted in different areas around the world (Jones et al., 2020; Lee et al., 2017; Liu et al., 2016; Liu et al., 2019; Lu et al., 2020; Miri et al., 2019; Ross et al., 2007; Wu et al., 2017). However, the development and application of LUR models in the Taiwan region have been limited (Hsu et al., 2019). In addition, most previous Taiwan LUR studies used data from purpose-designed monitoring networks or combined purpose-designed and routine monitoring networks (Ho et al., 2015; Lee et al., 2014; Lee et al., 2015). For example, Lee et al. (2015) established LUR models for ambient particles of aerodynamic diameter less than or equal to 2.5 μm ($PM_{2.5}$) using a purpose-designed monitoring network of 20 sites in the Taipei metropolis. The purpose-designed monitoring campaign has the advantage of capturing short-term air pollution exposure profiles (Jones et al., 2020), but it typically requires extra human labor and resources (e.g., experimental materials) (Hoek et al., 2008). Moreover, it is almost impossible to conduct long-term measurement (e.g., over years) using purpose-designed monitoring networks (Ho et al., 2015; Lee et al., 2017). As a result, a general limitation of LUR models upon purpose-designed monitoring networks is that the established models are usually only valid during the measurement period (Hoek et al., 2008; Shi et al., 2020). Therefore, the development of long-term





average LUR models for specific air pollutants using only routine monitoring networks should be explored, which is especially critical for epidemiological studies.

The application of established LUR models to areas outside the study area can reduce extra efforts to develop new models (Poplawski et al., 2009). To date, a few studies have evaluated the transferability of air pollution LUR models within a city and between cities or countries (Allen et al., 2011; Patton et al., 2015; Vienneau et al., 2010; Yang et al., 2020). Direct

transferability refers to predictor variables and coefficients of LUR models both being transferred (Allen et al., 2011), whereas transferability with calibration means that model coefficients are calibrated using air pollutant measurements from the target areas (Yang et al., 2020). Direct transferability is more meaningful because it can be applied in areas without air quality measurements (Allen et al., 2011; Yang et al., 2020). They concluded that the predictive performances of LUR models from one area to another were not consistent, ranging from poor (Marcon et al., 2015) to relatively acceptable

predictive accuracy (Poplawski et al., 2009; Wang et al., 2014). Therefore, more studies should be conducted to assess the transferability of air pollution LUR models.

In this study, annual average LUR models and spatial distribution maps were developed for ambient particles of aerodynamic diameter less than or equal to 10 μm ($PM_{10}$), $PM_{2.5}$, nitrogen dioxide ($NO_2$), and ozone ($O_3$) in northern Taiwan in 2019. In addition, the transferability of LUR models between cities in the study area was evaluated. The remainder of this

paper is organized as follows: the Materials and methods section describes the study area, data collection and processing, LUR model establishment and validation, and prediction of the air pollution exposure surface; the Results and discussion section presents an overview of measurement data, established LUR models and their comparison with previous LUR models in Taiwan, the transferability of LUR models, the spatial distribution maps of ambient $PM_{10}$, $PM_{2.5}$, $NO_2$, and $O_3$ concentrations, and $PM_{2.5}$ exposure estimates for a cohort study; and the Conclusions section summarizes the main results

and demonstrates the implications of the present study.

## 2 Materials and methods

### 2.1 Study area

The Taipei–Keelung metropolitan area (TKMA), located in northern Taiwan, includes Taipei City, New Taipei City, and Keelung City. The TKMA is the political, cultural, and social-economic center of Taiwan. It covers an area of approximately

2457 $km^2$, and has 48 administrative districts (Chiu et al., 2019; Wang et al., 2018). The TKMA had a population of about 7.03 million in 2019 (TWMOI, 2020), accounting for approximately 30% of the total population of Taiwan (Fig. 1(a)). The population densities of Taipei City, New Taipei City, and Keelung City were 10,175 people/$km^2$, 2021 people/$km^2$, and 2826 people/$km^2$, respectively, in 2019 (TWMOI, 2020). The numbers of registered motor vehicles were 1.76 million, 3.21 million, and 0.28 million in Taipei City, New Taipei City, and Keelung City, respectively, by the end of 2018 (TWMOTC,

2020).



The TKMA is situated in the subtropical region and on the downwind side of Mainland China. The built-up area of the TKMA is located in the central part of the Tamsui river basin surrounded by mountains, agricultural land, and forests (Fig. 1(b) & (c)). The characteristics of the basin terrain can constrain the diffusion of polluted air masses and thus favor the accumulation of air pollution in urban areas (Yu and Wang, 2010). Local emission sources of air pollutants in the TKMA
include vehicular exhaust, industrial emissions, and various sources related to residential activities (e.g., cooking) (Chen et al., 2020; Ho et al., 2018; Wu et al., 2017). In winter time, the long-distance transport of dust and polluted air masses under the northeast monsoon from the Asian continent results in a significant increase in concentrations of air pollutants (Chi et al., 2017; Chou et al., 2010).

## 2.2 Data collection and processing

The Taiwan Environmental Protection Administration (TWEPA) operates 20 central air quality monitoring stations in the TKMA, of which 12 stations are in New Taipei City, 7 are in Taipei City, and 1 station is in Keelung City (https://airtw.epa.gov.tw/ENG/default.aspx). In addition, the Taipei Environmental Protection Agency (TPEPA) operates 10 local air quality monitoring stations (https://www.tldep.gov.taipei/EIACEP_EN/Air_NormalStation.aspx). In total, these stations include 21 general stations, 6 traffic stations, 2 background stations, and 1 country park station (Fig. 1(a)). Detailed
descriptions of sampling stations, measurement instruments, and quality assurance and control procedures are available in TWEPA (2020). Hourly measurements of ambient $PM_{10}$, $PM_{2.5}$, $NO_2$, and $O_3$ concentrations and the meteorological variables of temperature, wind speed, and relative humidity at the central stations from January 01, 2019 to December 31, 2019 were collected from the Environment Resource database of TWEPA (https://erdb.epa.gov.tw/DataRepository/EnvMonitor/AirQualityMonitorDayData.aspx). In addition, hourly concentrations
of ambient $PM_{10}$, $PM_{2.5}$, $NO_2$, and $O_3$ at the local stations from January 01, 2019 to December 31, 2019 were downloaded from the TPEPA website (https://www.tldep.gov.taipei/Public/DownLoad/AirAutoHour.aspx). We calculated daily average values of air pollutant concentrations and meteorological variables from hourly data, and calculated the annual average values from daily averaged data for the development of LUR models. Daily and annual average estimates for the air pollutants require at least 75% data completeness (Cai et al., 2020); otherwise there is no value estimate for that day or year.

As presented in Table S1 and Fig. 1, the potential predictor variables of the road network, land use data, normalized difference vegetation index (NDVI), population density, and digital elevation data, which were frequently used in previous LUR studies, were collected. Land-use information was taken from the Land Use Investigation of Taiwan conducted by the National Land Surveying and Mapping Center (https://www.nlsc.gov.tw/LUI/Home/Content_Home.aspx). The Taiwan land-use status is classified into 9 main categories, 41 subcategories, and 103 detailed items. As shown in Fig. 1(c), the 9 main
land-use categories are agriculture, forest, transportation, water bodies, built-up areas, public utilities, recreation, mining or salt production, and others (Chen et al., 2020). The road network from the Taiwan Ministry of Transportation and Communications includes three types of road: local roads, major roads, and expressways (Fig. 1(d)). The NDVI and





elevation data were extracted from the database of the Resources and Environmental Sciences Data Center, Chinese Academy of Sciences (http://www.resdc.cn).

The values of potential predictor variables in buffer sizes of 50 m, 100 m, 300 m, 500 m, 700 m, 1000 m, 2000 m, 3000 m, 4000 m, and 5000 m surrounding the sampling stations were summarized for use in LUR model development. To ensure the consistency of results between model training and cross validation, we included only the potential predictor variables with at least 7 stations (i.e., around 25% of all stations) exhibiting different values and where the minimum or maximum values lay within three times the 10th to the 90th percentile range below or above the 10th and the 90th percentile (Wolf et al., 2017).

## 2.3 Model development and validation

The LUR models of ambient $PM_{10}$, $PM_{2.5}$, $NO_2$, and $O_3$ for the entire study area (the area-specific LUR models) were established using all 30 air quality monitoring stations. In addition, city-specific LUR models for New Taipei & Keelung City were developed using the 13 quality monitoring stations located in these two cities, and the established models were directly transferred to Taipei City. Similarly, city-specific LUR models for Taipei City were developed using the 17 quality

monitoring stations located in this city, and the established models were directly transferred to New Taipei & Keelung City. In this study, we did not consider the calibration of model coefficients because we planned to evaluate the direct transferability of city-specific LUR models to another nearby city area when there were no routine air quality measurements.

There is no standard modeling method for developing LUR models (Hoek et al., 2008). In this study, the supervised forward linear regression method (Cai et al., 2020; Eeftens et al., 2016; Xu et al., 2019) was used to develop the LUR models. This

modeling method can ensure that only predictor variables following the plausible direction of effect are included and meanwhile the predictive accuracy of the established model is maximized. In brief, all potential predictor variables were included as candidate independent variables and a prior direction was assigned for each category of variable based on the atmospheric mechanism. The model construction started by including the predictor variable with the highest adjusted explained variance ($R^2$). The remaining predictor variables were entered into the model if they met all of the following

criteria: 1) the gain of the adjusted $R^2$ was no less than 1%; 2) the direction of effect of the predictor variable was pre-defined; 3) variables were added into the model when the probability of $F$ was less than 0.05 and removed when the probability of $F$ was greater than 0.10; 4) variables already included in the model retained the same direction of effect; and 5) following previous studies (Chen et al., 2020; Marcon et al., 2015; Wang et al., 2014), the predictor variables with variance inflation factor (VIF) values larger than 3 were dropped to make a tradeoff between model interpretation and the

predictive accuracy (Eeftens et al., 2016). Multiple buffer sizes of a specific variable (e.g., the length of local roads) could be selected in the final model as long as they followed the selection criteria (Henderson et al., 2007).

Standard diagnostic tests were applied to ensure that the LUR models were reasonably established (Li, 2020; Wolf et al., 2017). The Cook's distance value was calculated to detect the outliers of data points (i.e., stations) (Jones et al., 2020). Air pollutant observations with a Cook's distance value greater than 1 would be excluded and the LUR model for this air





pollutant would be re-established (Weissert et al., 2018; Wolf et al., 2017). In addition, Moran's *I* values on the concentrations residuals of the final LUR models were calculated using ArcGIS software to evaluate the spatial autocorrelation (Bertazzon et al., 2015; Lee et al., 2017; Liu et al., 2016). The $R^2$ and root mean square error (RMSE) were estimated to evaluate the performance of the models. Furthermore, leave-one-out cross validation (LOOCV) was employed to evaluate the predictive capacity of the LUR models (Liu et al., 2019; Shi et al., 2020; Yang et al., 2020).

Spatial analysis and calculations were performed using ArcGIS software, version 10.6 (ESRI Inc., Redlands, CA, USA). The statistical analysis was performed using R software, version 3.5.2 (R Core Team, 2018).

## 2.4 Air pollution surface prediction

The entire study area of the TKMA was divided into 9839 500 m × 500 m grid cells. The air pollutant concentrations at the centroids of the grid cells were estimated using the established area-specific LUR models. When the LUR models estimated

negative concentration values, the concentration values of the grid cells were set to zero; when air pollutant concentration estimates exceeded the maximum observed concentrations by more than 20%, the concentrations of grid cells were set to 120% of the maximum observed concentrations (Henderson et al., 2007). The area-specific LUR model-based negative and high concentration estimates accounted for only 0%, 4%, 2%, and 0% of $PM_{10}$, $PM_{2.5}$, $NO_2$, and $O_3$ estimates, respectively. Then the spatial distribution maps of ambient $PM_{10}$, $PM_{2.5}$, $NO_2$, and $O_3$ concentrations were created using the kriging

interpolation method (Cai et al., 2020).

## 3 Results and discussion

### 3.1 Descriptive statistics of the air quality data

In general, the included air quality monitoring stations were situated at different types of land uses across the TKMA (Table 1 and Fig. 1(c)), which suggests that the collected data set has relatively good representativeness. The annual average $PM_{10}$

concentration of 39.3 μg/m³ at background stations was the highest, followed in descending order by traffic stations with 33.6 μg/m³, general stations with 28.5 μg/m³, and the country park station with 15.7 μg/m³. The traffic stations and country park station had the highest and lowest annual average $PM_{2.5}$ concentrations, respectively. The annual average $PM_{2.5}$ concentrations at general stations of 13.7 μg/m³ and background stations of 13.2 μg/m³ were comparable. Except for the country park station, the annual average $PM_{10}$ and $PM_{2.5}$ concentrations at other types of stations were higher than the air

quality guidelines (AQGs) for $PM_{10}$ and $PM_{2.5}$ of 20 μg/m³ and 10 μg/m³, respectively, proposed by the World Health Organization (WHO) (WHO, 2006). The annual average $NO_2$ concentration of 24.6 ppb at the traffic stations was the highest, followed by general stations with 14.3 ppb. The annual average $NO_2$ concentrations at background stations (3.81 ppb) and the country park station (1.89 ppb) were significantly lower than those of general and traffic stations because they were farther away from traffic emissions. The annual average $NO_2$ concentration at traffic stations (24.6 ppb) was slightly





higher than the WHO $NO_2$ AQG of 40 μg/m³ (about 21.3 ppb) (WHO, 2006), while other types of stations had annual

average $NO_2$ concentrations lower than this AQG. In contrast to $NO_2$, the background stations (41.7 ppb) and the country

park station (39.8 ppb) had higher annual average $O_3$ concentrations than those of traffic stations (21.6 ppb) or general

stations (29.4 ppb) (Table 1).

**3.2 The area-specific LUR models**

Fig. S1 shows that Cook's distance values were below 1 for all the stations of the area-specific LUR models, suggesting that

there were no station outliers in developing these LUR models. For $PM_{10}$ and $PM_{2.5}$ LUR models, Cook's distance values

ranged from almost 0.00 to around 0.72. The Cook's distance values of the $NO_2$ LUR model were between almost 0.00 and

0.28, whereas the Cook's distance values of the $O_3$ LUR model were between almost 0.00 and 0.38 (Fig. S1). The final area-

specific LUR models and their corresponding predictive accuracy are summarized in Table 2 and Fig. 2. The model $R^2$

values ranged from 0.72 for $PM_{2.5}$ to 0.91 for $NO_2$, indicating a good fit for all air pollutants. $PM_{10}$, $NO_2$, and $O_3$ LUR

models performed well, with LOOCV $R^2$ values being < 0.10 lower than the model $R^2$ values. For $PM_{2.5}$, the model was not

as robust as those of other air pollutants, with the LOOCV $R^2$ value being 0.19 lower than the model $R^2$ value (Fig. 2). The

reason for this is that the $PM_{2.5}$ concentrations among the stations were not as discrete as those of other air pollutants (Table

1 and Fig. 2). The significance of the predictor variables (*p* value) and VIF values all met the requirements for LUR model

development. Moran's *I* values were 0.0047, −0.072, 0.023, and −0.055 for the LUR models of ambient $PM_{10}$, $PM_{2.5}$, $NO_2$,

and $O_3$. In addition, z-score values were 0.83, −0.79, 1.2, and −0.34 for ambient $PM_{10}$, $PM_{2.5}$, $NO_2$, and $O_3$ LUR models,

respectively, indicating that the spatial patterns of concentration residuals of the LUR models do not appear to be

significantly different from random (Fig. S2).

The final area-specific LUR models consisted of three (for $O_3$), four (for $NO_2$), and five predictor variables (for $PM_{10}$ and

$PM_{2.5}$) (Table 2). Consistent with the previous LUR studies of De Hoogh et al. (2018), Eeftens et al. (2016), Jones et al.

(2020), Weissert et al. (2018) and Wolf et al. (2017), the established LUR models contained at least one traffic-related

predictor variable in buffer sizes ranging from 50 m to 3000 m. Traffic emission is a major source of air pollution in urban

areas of the TKMA (Lee et al., 2014; Wu et al., 2017). For instance, it was reported that gasoline and diesel vehicle

emissions contributed approximately half of $PM_{2.5}$ concentrations in Taipei City based on source apportionment analysis (Ho

et al., 2018). Several previous LUR studies selected the population density variable as the final explanatory variable in their

$PM_{2.5}$ and $NO_2$ LUR models (Ji et al., 2019; Meng et al., 2015; Rahman et al., 2017). However, it was not included in our

final LUR models. A possible explanation is that the population density variable is moderately or highly correlated with the

variables (e.g., the area of recreational land) included in our final LUR models.

As shown in Table 2, $PM_{10}$ and $PM_{2.5}$ LUR models included predictor variables of both small and large buffer sizes. The

LUR model for $PM_{10}$ included the area of forest land in a buffer size of 300 m, the area of built-up land in a buffer size of 50

m, the area of recreational land in a buffer size of 2000 m, the area of transportation land in a buffer size of 100 m, and the



area of waterbody land in a buffer size of 500 m. The PM$_{2.5}$ LUR model included the area of transportation land within a 300-m buffer, the area of major roads within a 100-m buffer, the area of forest land within a 700-m buffer, the area of recreational land within a 2000-m buffer, and the distance to the nearest major roads. For PM$_{10}$ and PM$_{2.5}$ LUR models, the

direction of effect for transportation land and traffic roads was positive, while the direction of effect of other predictor variables was negative. Forest and urban green space land (i.e., recreational land) were included in both PM$_{10}$ and PM$_{2.5}$ LUR models (Table 2). Ji et al. (2019), Jones et al. (2020), and Miri et al. (2020) included forest land or urban green space as the predictor variables in their final city-scale PM LUR models, demonstrating the mitigation effect of these land-use types on PM concentrations. Chen et al. (2019) and Jeanjean et al. (2016) reported the effectiveness of urban green space in

mitigating PM pollution. The waterbody type of land-use reduced PM$_{10}$ concentrations, as evidenced by the negative regression coefficient (Table 2). The waterbodies can make PM$_{10}$ absorb moisture and increase sedimentation. In addition, large areas of water provide good conditions for the dispersion of air pollutants (Zhu and Zhou, 2019).

For the NO$_2$ LUR model, the four predictor variables included were the area of transportation land in buffer sizes of 3000 m and 50 m, the area of recreational land in a 1000-m buffer, and the sum of the length of local roads in a 1000-m buffer. The

direction of effect for the recreational land was negative, while other predictor variables showed a positive effect (Table 2). The O$_3$ LUR model included predictor variables with relatively small buffer sizes of less than 700 m. The three predictor variables were the area of transportation land in buffer sizes of 700 m and 50 m, and the area of public utilization land within a 300-m buffer. The directions of effect for these three variables were all negative (Table 2). The traffic-related predictor variables were important variables in predicting NO$_2$ and O$_3$ concentrations but in different directions of effect. Consistent

with previous studies by De Hoogh et al. (2016), Eeftens et al. (2016), Lee et al. (2014), and Liu et al. (2019), the established NO$_2$ LUR model also revealed the mitigation effect of urban green space (i.e., recreational land) on NO$_2$ concentration.

A comparison of this study with previous LUR studies in Taiwan is presented in Table S2. The predictive performance of the LUR model for ambient PM$_{10}$ in this study was slightly worse than that of Lee et al. (2015) with an $R^2$ value of 0.87. In addition, the $R^2$ and LOOCV $R^2$ values (0.72 and 0.53, respectively) of the PM$_{2.5}$ LUR model in this study were lower than

those of Ho et al. (2015) (an $R^2$ value of 0.75 and an LOOCV $R^2$ value of 0.62), Lee et al. (2015) (an $R^2$ value of 0.95 and an LOOCV $R^2$ value of 0.91), and Wu et al. (2017) (an $R^2$ value of 0.90 and an LOOCV $R^2$ value of 0.83), but higher than that of Wu et al. (2018) with an $R^2$ value of 0.66. The NO$_2$ LUR model performed better than that of Lee et al. (2014) and was comparable to that of Chen et al. (2020). Hsu et al. (2019) developed an O$_3$ LUR model for the whole of Taiwan region, with an $R^2$ value of 0.74 (Hsu et al., 2019). Our study established a reasonable LUR model for ambient O$_3$ in the TKMA with an

$R^2$ value of 0.80 and an LOOCV $R^2$ value of 0.70, which is a relatively high predictive performance. Compared with PM$_{10}$, PM$_{2.5}$, and NO$_2$, the establishment of O$_3$ LUR models has been limited in these previous Taiwan LUR studies (Table S2) or in most of the LUR studies in other areas, but it is essential to establish O$_3$ LUR models given that O$_3$ is a toxic photochemical pollutant threatening human health and the ecosystem (Ning et al., 2020; Yim et al., 2019).



### 3.3 Transferability of the city-specific LUR models

The city-specific LUR models for ambient $PM_{10}$, $PM_{2.5}$, $NO_2$, and $O_3$ in Taipei City and New Taipei & Keelung City are shown in Tables S3 and S4, respectively. The model $R^2$ values of the Taipei City $PM_{10}$, $PM_{2.5}$, $NO_2$, and $O_3$ LUR models were 0.91, 0.64, 0.89, and 0.76, respectively (Tables S3), while the New Taipei City & Keelung City $PM_{10}$, $PM_{2.5}$, $NO_2$, and $O_3$ LUR models had $R^2$ values of 0.63, 0.65, 0.95, and 0.93, respectively (Tables S4). In general, for each specific air pollutant, the predictive performance of these city-specific LUR models can be slightly higher or lower than those of the

area-specific LUR models. Fig. 3 shows the transferability of LUR models between Taipei City and New Taipei & Keelung City. The city-specific LUR models performed worse in another city area than in the city where these models were established. For instance, the transfer-predictive $R^2$ values of the Taipei LUR models were 0.31, 0.04, 0.62, and 0.56 for predicting ambient $PM_{10}$, $PM_{2.5}$, $NO_2$, and $O_3$ in New Taipei & Keelung City, respectively (Fig. 3). These values were substantially lower than the corresponding $R^2$ values of the Taipei LUR models. The $NO_2$ LUR models showed good

transferability between the two city areas, with transfer-predictive $R^2$ values higher than 0.62. However, the $PM_{10}$, $PM_{2.5}$, and $O_3$ LUR models performed poorly when they were transferred between the two city areas, with transfer-predictive $R^2$ values of $< 0.31$, $< 0.37$ and $< 0.56$, respectively (Fig. 3). Similar to the previous studies of Marcon et al. (2015) and Yang et al. (2020), these results suggested that there may be large uncertainties in transferring LUR models between cities, and even between nearby cities with similar geographic and urban design characteristics. The use of novel cost-effective methods

(e.g., low-cost air quality sensors or satellite remote sensing approach) is therefore recommended to assess air pollution and associated population exposure in cities with limited fixed-site measurement stations.

### 3.4 Spatial maps

LUR-model-derived air pollution spatial distribution maps provide valuable and useful air pollutant concentration surfaces in the TKMA. In general, there was a good agreement between LUR-model-based concentration estimates and observations for

$PM_{10}$, $PM_{2.5}$, $NO_2$, and $O_3$ (Fig. 4). For $PM_{10}$ and $PM_{2.5}$, there were certain differences between LUR-model-based concentration estimates and observations at the country park station (Fig. 4). A possible reason for this difference may be that the kriging interpolation method removed low-concentration estimates at this small area when the concentration estimates at nearby areas were higher.

High concentrations of ambient $PM_{10}$, $PM_{2.5}$, and $NO_2$ were predicted in the urban areas of Taipei City, New Taipei City, and

Keelung City, and along the road network. The estimated $PM_{10}$ and $PM_{2.5}$ concentrations in urban areas were around 35.0 to 40.9 $\mu g/m^3$ and around 12.0 to 17.0 $\mu g/m^3$, respectively, whereas the urban areas had $NO_2$ concentrations of around 12.0 to 31.7 ppb (Fig. 4). This spatial distribution pattern is understandable given that the traffic-related predictor variables were included in the final $PM_{10}$, $PM_{2.5}$, and $NO_2$ LUR models. A similar spatial pattern of $PM_{2.5}$ concentrations was reported by Wu et al. (2017), which documented that high $PM_{2.5}$ concentrations were distributed mainly in the urban areas of the TKMA

and there were also scattered points of high $PM_{2.5}$ concentrations in its outer ring. However, the estimated 2019 annual



average PM$_{2.5}$ concentrations in this study were significantly lower than those for 2006−2012 estimated by Wu et al. (2017). There was a clear decreasing trend in PM$_{2.5}$ concentrations in the whole of Taiwan over the past decade (Ho et al., 2020; Jung et al., 2018). For example, Jung et al. (2018) reported that the estimated PM$_{2.5}$ concentrations declined by 1.7 µg/m$^3$ and 1.6 µg/m$^3$ in the morning and afternoon, respectively, per year over the whole of Taiwan during the period 2005−2015.

O$_3$ showed a generally opposite spatial variability pattern compared with the other three air pollutants, with lower concentrations (< about 32.0 ppb) in urban areas than in rural areas (Fig. 4). A possible explanation for this finding is that high concentrations of NO and NO$_2$ in urban areas react with O$_3$, resulting in a decrease in O$_3$ concentration (Hsu et al., 2019; Vardoulakis et al., 2011).

Correlations of estimated concentrations of PM$_{10}$, PM$_{2.5}$, NO$_2$, and O$_3$ in the TKMA are shown in Table 3. Consistent with

previous studies by Hoek et al. (2008), Lu et al. (2020), Vardoulakis et al. (2011), and Wolf et al. (2017), the spatial distribution maps revealed high spatial correlations among the four air pollutants. PM$_{10}$ concentrations had strong positive correlations with PM$_{2.5}$ and NO$_2$, suggesting common sources of these three air pollutants. In contrast to this, PM$_{10}$ concentrations were negatively correlated with O$_3$ concentrations, with a Pearson correlation coefficient (PCC) value of −0.730. Similarly, PM$_{2.5}$ concentrations had a strong positive correlation with NO$_2$ concentrations but showed a significant

negative correlation with O$_3$ concentrations. The concentrations of NO$_2$ and O$_3$ were negatively correlated, with a PCC value of −0.920. Similar findings were reported by De Hoogh et al. (2018) and Lu et al. (2020).

### 3.5 Air pollutant exposure estimates for a cohort study

Air pollutant concentrations measured at nearby fixed-site stations are often used to represent exposures in epidemiological studies (Lin et al., 2016; Shi et al., 2020), but the spatial resolution of these estimates is relatively coarse due to the limited

number of sampling stations (Bertazzon et al., 2015). In recent years, LUR modeling has become a more widely applied method to estimate air pollution exposures at a fine spatial scale (Lee et al., 2014; Wolf et al., 2017). Fig. S3 shows that there are differences between LUR-model-based air pollution exposure estimates and nearby-station measurements at residential locations of participants in a cohort study conducted in the TKMA. The average values of the LUR-estimated PM$_{10}$, PM$_{2.5}$, NO$_2$, and O$_3$ exposure concentrations were 36.0 µg/m$^3$, 14.2 µg/m$^3$, 18.0 ppb, and 29.2 ppb, respectively, whereas the

corresponding nearby-station measurements were 27.7 µg/m$^3$, 13.8 µg/m$^3$, 16.3 ppb, and 28.6 ppb, respectively (Table S5). Compared with LUR-model-based estimates, the nearby-station measurements underestimated PM$_{10}$, PM$_{2.5}$, NO$_2$, and O$_3$ exposures of cohort participants by 8.23 µg/m$^3$, 0.41 µg/m$^3$, 1.73 ppb, and 0.60 ppb, respectively (Table S5). In addition, the concentration ranges of LUR-estimated annual average PM$_{10}$ (13.0–45.2 µg/m$^3$), PM$_{2.5}$ (6.96–19.9 µg/m$^3$), NO$_2$ (0.70–32.2 ppb), and O$_3$ (17.5–44.0 ppb) exposure concentrations were larger than those of nearby-station measurements for PM$_{10}$

(22.3–40.3 µg/m$^3$), PM$_{2.5}$ (10.6–21.3 µg/m$^3$), NO$_2$ (2.90–32.2 ppb), and O$_3$ (15.2–42.2 ppb) (Table S5 and Fig. 5). This indicates that the LUR-model-based exposure estimates can capture the large spatial variability in air pollutant exposure among the cohort participants. Furthermore, the LUR-model-based PM$_{10}$, PM$_{2.5}$, NO$_2$, and O$_3$ exposure estimates and


nearby-station measurements were weakly correlated, with linear regression $R^2$ values ranging from 0.05 for $PM_{10}$ to 0.19 for $NO_2$ (Fig. 6). The obtained results highlight that air pollution LUR models may provide more accurate exposure estimates

than nearby-station measurements.

### 3.6 Limitations

This study is subject to several limitations. First, apart from the variables used in this study, more predictor variables (e.g., localized emission data and urban building morphology data) should be included and tested to develop LUR models. For example, Wu et al. (2017) and Chen et al. (2020) assessed the roles of two culturally specific emission sources, Chinese

restaurants and temples, on the development of ambient $PM_{2.5}$ and $NO_2$ LUR models in Taiwan. More studies should be conducted to test the influence of different potential predictor variables on the development of LUR models (Hoek et al., 2008). Second, like most linear regression techniques, the supervised forward linear regression method is not proficient in modeling extreme values (Jones et al., 2020). In addition, there may be complex and non-linear relationships between the explanatory variables and air pollutant concentrations (Wang et al., 2020). Other types of linear regression methods (Hoek et

al., 2018; Shi et al, 2020) and the novel machine learning algorithms (Wang et al., 2020) can be tested in estimating surface-level air pollutant concentrations in the further study. Third, the kriging interpolation method tends to remove air pollutant peak concentrations, resulting in an underestimation of air pollution exposure at pollution hotspots. Other spatial mapping methods should be considered in further studies. It is recommended that air pollutant concentrations at residential locations of participants should be estimated directly for cohort studies. Fourth, there may be uncertainty in spatial estimations of air

pollutant concentrations with a limited number of sampling stations. Further studies are warranted to evaluate the influence of the number of sampling stations and their spatial distributions on the development of LUR models and the air pollution spatial maps.

### 4 Conclusions

Following standard development procedures, the annual average LUR models of ambient $PM_{10}$, $PM_{2.5}$, $NO_2$, and $O_3$ were

established in the TKMA of northern Taiwan using only data from the routine monitoring network. These LUR models were reasonable, based on the evaluation metrics of Cook's distance, VIF, Moran's $I$, and $p$ values. The $R^2$ values of the LUR models for ambient $PM_{10}$, $PM_{2.5}$, $NO_2$, and $O_3$ were 0.80, 0.72, 0.91, and 0.80, respectively. The traffic-related predictor variables were the major explanatory factors in the LUR models for all the studied air pollutants.

The predictive performance varied greatly among air pollutants in examining the transferability of city-specific LUR models

between New Taipei & Keelung City and Taipei City, with relatively high transfer-predictive $R^2$ values for $NO_2$. Therefore, this study highlights that the established LUR models in a city area can result in a large estimation bias when applied to another nearby city area with similar geographic and urbanization conditions. It is necessary to conduct more studies to evaluate and improve the intercity transferability of LUR models.



The spatial distribution maps of the four air pollutants showed that the developed LUR models are reasonable in modeling the spatial variabilities of air pollution. Ambient $PM_{10}$, $PM_{2.5}$, and $NO_2$ shared similar spatial variations, with relatively high concentrations in urban areas and along the road network. Ambient $O_3$ presented a generally opposite spatial variability compared with $PM_{10}$, $PM_{2.5}$, or $NO_2$. These estimated air pollution concentration surfaces provide information for the management of air pollution and exposure estimates for epidemiological studies. Compared with nearby-station measurements, the LUR-model-based concentration estimates captured a wider range of exposure to $PM_{10}$, $PM_{2.5}$, $NO_2$, and $O_3$ for participants in a cohort study in the TKMA. Further studies should pay more attention to utilizing other data sources (e.g., satellite remote sensing data) with comprehensive spatiotemporal coverage to validate the LUR-model-based estimations of air pollutant concentrations.

*Data availability.* The model data presented in this article are available from the authors upon request (steveyim@cuhk.edu.hk).

*Author contributions.* SHLY planned, supervised and sought funding for this study. ZYL performed the data analysis and prepared the paper with contributions from all co-authors.

*Competing interests.* The authors declare that they have no conflict of interest.

*Acknowledgements.* We would like to thank the Taiwan Environmental Protection Administration for providing air quality and meteorological data.

*Financial support.* This work is funded by the Vice-Chancellor's Discretionary Fund of The Chinese University of Hong Kong (grant no. 4930744), the project from ENvironmental SUstainability and REsilience (ENSURE) partnership between CUHK and UoE ( http://www.exeter.ac.uk/cuhkpartnership/) and Dr. Stanley Ho Medicine Development Foundation (grant no. 8305509).

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





**Table 1** Statistical description of measured air pollutants by different types of stations.

| Air pollutant | Station type | N | Mean | SD | Min | Max |
|---|---|---|---|---|---|---|
| PM$_{10}$ (μg/m$^3$) | General | 21 | 28.5 | 2.84 | 22.3 | 35.0 |
| | Traffic | 6 | 33.6 | 4.57 | 27.3 | 40.3 |
| | Background | 2 | 39.3 | 2.11 | 37.8 | 40.8 |
| | Country park* | 1 | 15.7 | - | - | - |
| PM$_{2.5}$ (μg/m$^3$) | General | 21 | 13.7 | 1.36 | 10.6 | 15.4 |
| | Traffic | 6 | 16.8 | 2.98 | 13.3 | 21.3 |
| | Background | 2 | 13.2 | 0.44 | 12.9 | 13.6 |
| | Country park | 1 | 8.06 | - | - | - |
| NO$_2$ (ppb) | General | 21 | 14.3 | 3.32 | 7.86 | 21.7 |
| | Traffic | 6 | 24.6 | 6.16 | 17.1 | 32.2 |
| | Background | 2 | 3.81 | 1.28 | 2.90 | 4.71 |
| | Country park | 1 | 1.89 | - | - | - |
| O$_3$ (ppb) | General | 21 | 29.4 | 3.51 | 23.6 | 35.5 |
| | Traffic | 4 | 21.6 | 5.48 | 15.2 | 28.0 |
| | Background | 2 | 41.7 | 0.70 | 41.2 | 42.2 |
| | Country park | 1 | 39.8 | - | - | - |

Note: N means the number of stations for this type; SD means the standard deviation; Min and Max refer to the minimum and maximum values of the air pollutant concentrations, respectively. *only one country park station, therefore there are no estimates of SD, Min, and Max values.





**Table 2** Description of the 2019 annual average LUR models for ambient $PM_{10}$, $PM_{2.5}$, $NO_2$, and $O_3$ in the TKMA.

| Air pollutant | Variables | Coefficient | Standard error | $p$ | VIF | Predictive accuracy |
|---|---|---|---|---|---|---|
| **$PM_{10}$** | (constant) | 38.5 | 1.4 | < 0.001 | NA | $R^2$ = 0.80; RMSE = 2.25; LOOCV $R^2$ = 0.72; LOOCV RMSE = 2.83. |
| | LU2_300 | -7.71E-05 | 1.20E-05 | < 0.001 | 1.4 | |
| | LU5_50 | 1.01E-03 | 4.39E-04 | 0.031 | 1.3 | |
| | LU7_2000 | -7.06E-06 | 1.29E-06 | < 0.001 | 1.8 | |
| | LU3_100 | 5.33E-04 | 1.19E-04 | < 0.001 | 1.7 | |
| | LU4_500 | -2.97E-05 | 9.82E-06 | 0.006 | 1.1 | |
| **$PM_{2.5}$** | (constant) | 13.7 | 1.0 | < 0.001 | NA | $R^2$ = 0.72; RMSE = 1.25; LOOCV $R^2$ = 0.53; LOOCV RMSE = 1.69. |
| | LU3_300 | 4.26E-05 | 1.25E-05 | 0.002 | 1.7 | |
| | R2_100 | 3.52E-04 | 1.05E-04 | 0.003 | 1.2 | |
| | LU2_700 | -4.65E-06 | 1.34E-06 | 0.002 | 1.6 | |
| | LU7_2000 | -2.20E-06 | 8.03E-07 | 0.012 | 2.2 | |
| | Dis_Major | -5.70E+01 | 2.69E+01 | 0.045 | 1.1 | |
| **$NO_2$** | (constant) | 0.70 | 1.21 | 0.57 | NA | $R^2$ = 0.91; RMSE = 2.01; LOOCV $R^2$ = 0.88; LOOCV RMSE = 2.40. |
| | LU3_3000 | 1.77E-06 | 2.80E-07 | < 0.001 | 2.4 | |
| | LU3_50 | 2.35E-03 | 2.68E-04 | < 0.001 | 1.3 | |
| | LU7_1000 | -1.88E-05 | 3.30E-06 | < 0.001 | 1.5 | |
| | RL3_1000 | 4.91E-05 | 1.55E-05 | 0.004 | 2.0 | |
| **$O_3$** | (constant) | 44.0 | 1.7 | < 0.001 | NA | $R^2$ = 0.80; RMSE = 2.64; LOOCV $R^2$ = 0.72; LOOCV RMSE = 3.15. |
| | LU3_700 | -2.88E-05 | 4.00E-06 | < 0.001 | 1.1 | |
| | LU3_50 | -2.00E-03 | 3.65E-04 | < 0.001 | 1.1 | |
| | LU6_300 | -3.07E-05 | 1.20E-05 | 0.018 | 1.0 | |

**Note:**
LU2_300, LU2_700: the area of forest in buffer sizes of 300 m and 700 m
LU5_50: the area of built-up land in a buffer size of 50 m
LU7_1000 and LU7_2000: the area of recreational land in buffer sizes of 1000 m and 2000 m
LU3_50, LU3_100, LU3_300, LU3_700, and LU3_3000: the area of transportation land in buffer sizes of 50 m, 100 m, 300 m, 700 m, and 3000 m
LU4_500: the area of waterbody in a buffer size of 500 m
R2_100: the area of major roads in a buffer size of 100 m
Dis_Major: the distance to the nearest major roads
RL3_1000: the length of local roads in a buffer size of 1000 m
LU6_300: the area of public utilization land in a buffer size of 300 m
VIF: the variance inflation factor
LOOCV: leave-one-out cross validation
RMSE: root mean square error
NA: not available





**Table 3** Pearson correlation coefficients (PCCs) among the estimated concentrations of ambient $PM_{10}$, $PM_{2.5}$, $NO_2$, and $O_3$.

| Air pollutant | $PM_{10}$ | $PM_{2.5}$ | $NO_2$ | $O_3$ |
|---------------|-----------|------------|--------|-------|
| $PM_{10}$ | 1 | 0.775[**] | 0.719[**] | −0.730[**] |
| $PM_{2.5}$ | | 1 | 0.761[**] | −0.775[**] |
| $NO_2$ | | | 1 | −0.920[**] |
| $O_3$ | | | | 1 |

Note: ** Correlation is significant at the 0.01 level (2-tailed).





**Figure 1.** The characteristics of the study area. (a) Population density and the location of air quality monitoring stations. 30 air quality monitoring stations were included in this study. (b) Digital elevation. (c) Land use types. (d) The road network.





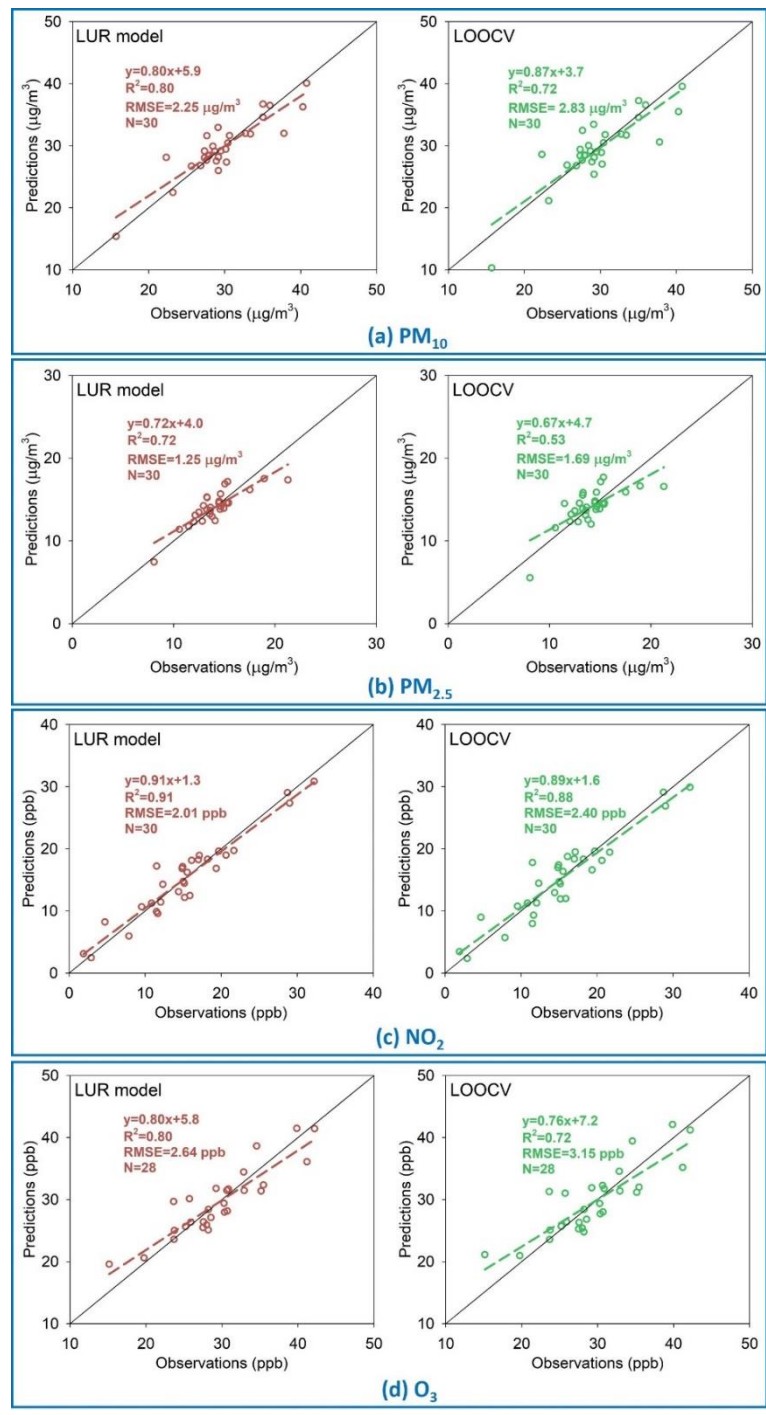

560

**Figure 2.** A comparison of LUR-predicted concentrations and observed concentrations of the studied air pollutants and the LOOCV-predicted concentrations and observed concentrations of the studied air pollutants. (a) $PM_{10}$, (b) $PM_{2.5}$, (c) $NO_2$, and (d) $O_3$. N is the sample size, and the solid line is the 1:1 line.





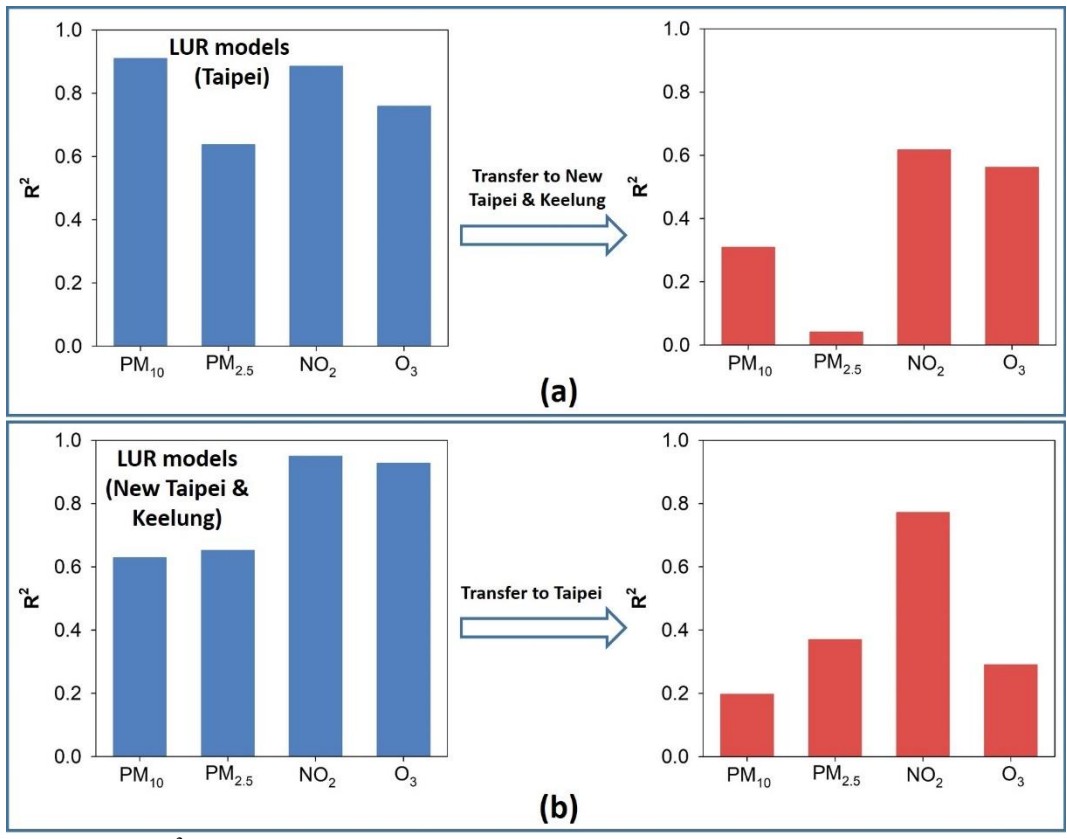

**Figure 3.** The changes in $R^2$ values for direct transfer of ambient $PM_{10}$, $PM_{2.5}$, $NO_2$, and $O_3$ LUR models between Taipei City and New Taipei & Keelung City.





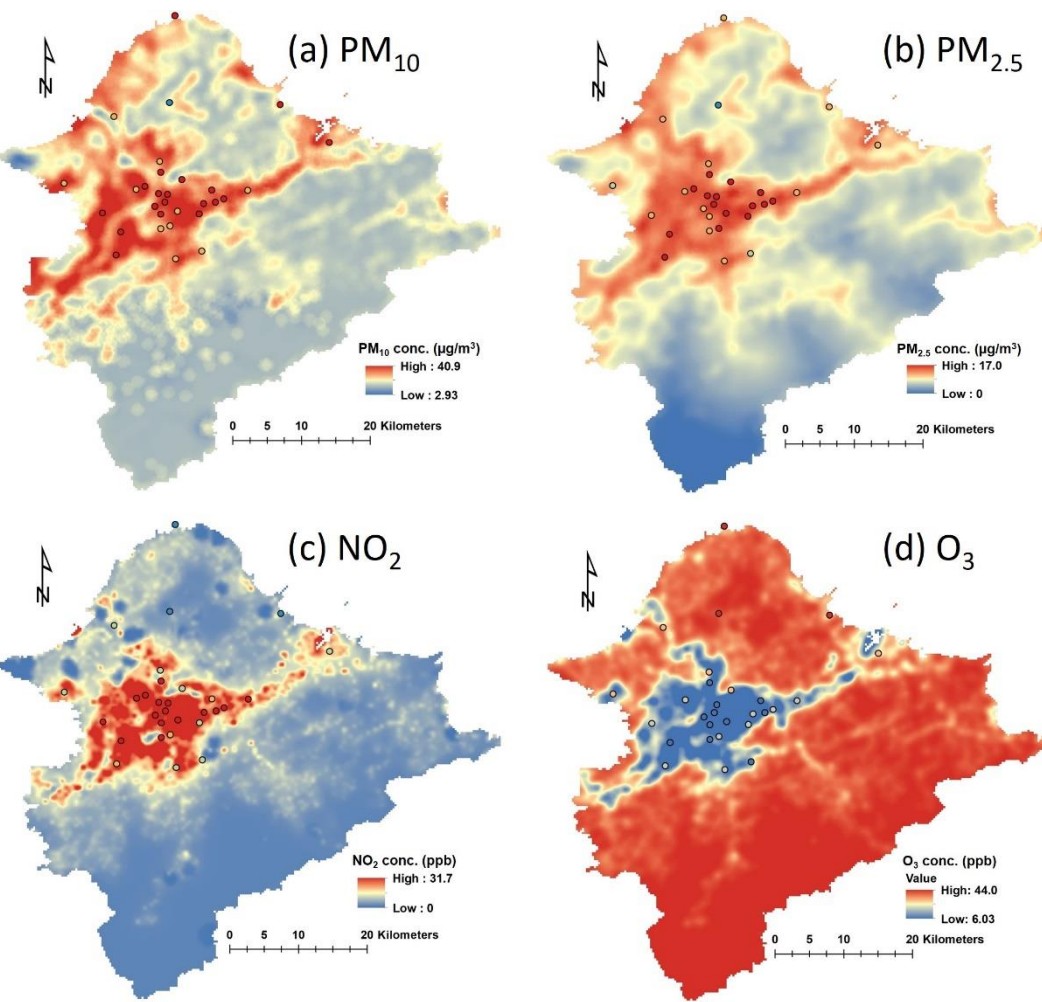

**Figure 4.** The spatial distribution of ambient air pollutant concentrations derived from established LUR models. (a) $PM_{10}$, (b) $PM_{2.5}$, (c) $NO_2$, and (d) $O_3$. The colored circles represent the observations from stations.

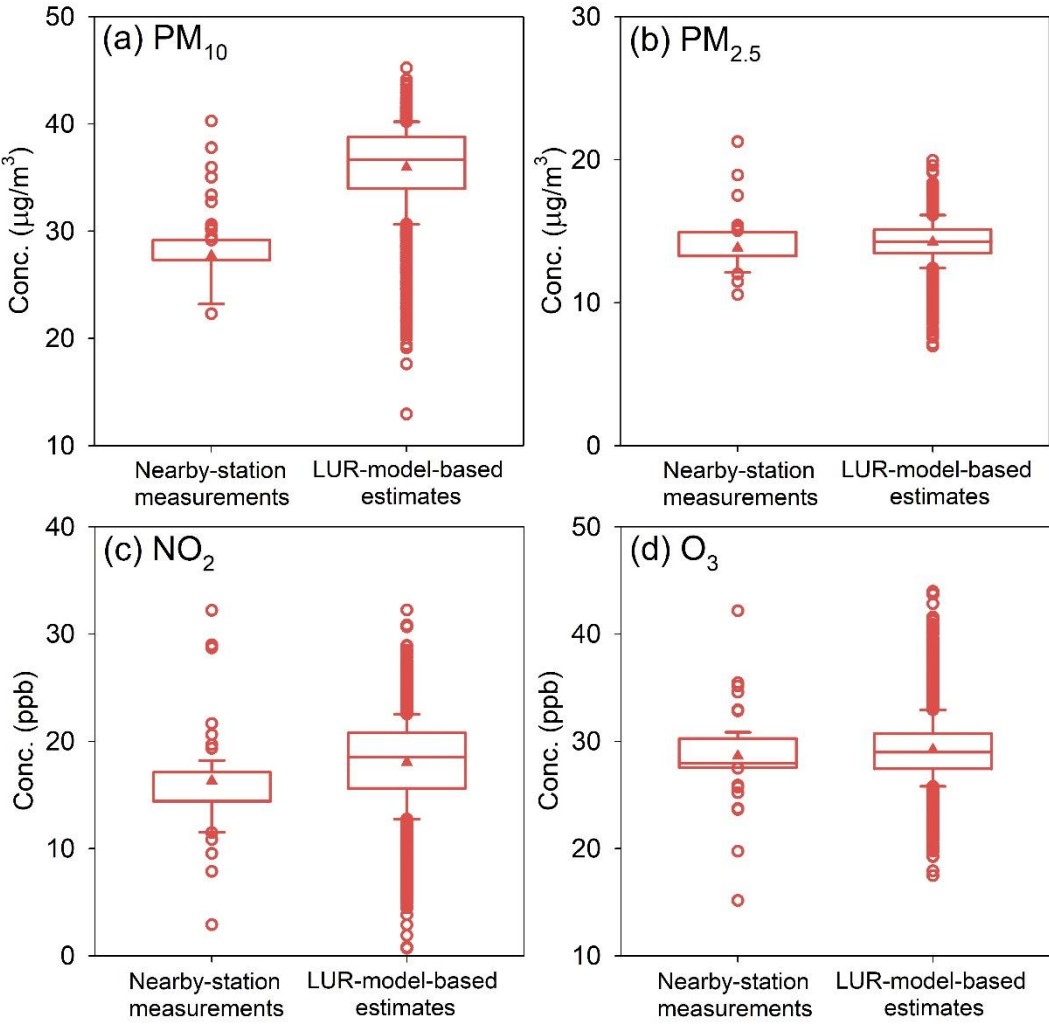

570

**Figure 5.** Box plots of nearby-station air pollutant measurements and LUR-model-based estimates of air pollutant concentration. (a) $PM_{10}$, (b) $PM_{2.5}$, (c) $NO_2$, and (d) $O_3$. The triangle symbol in each box is the mean value, the solid line is the median value, the box extends from the 25th to the 75th percentile, the whiskers (error bars) below and above the box are the 10th and 90th percentiles, and the lower and upper cycle symbols are outliers.





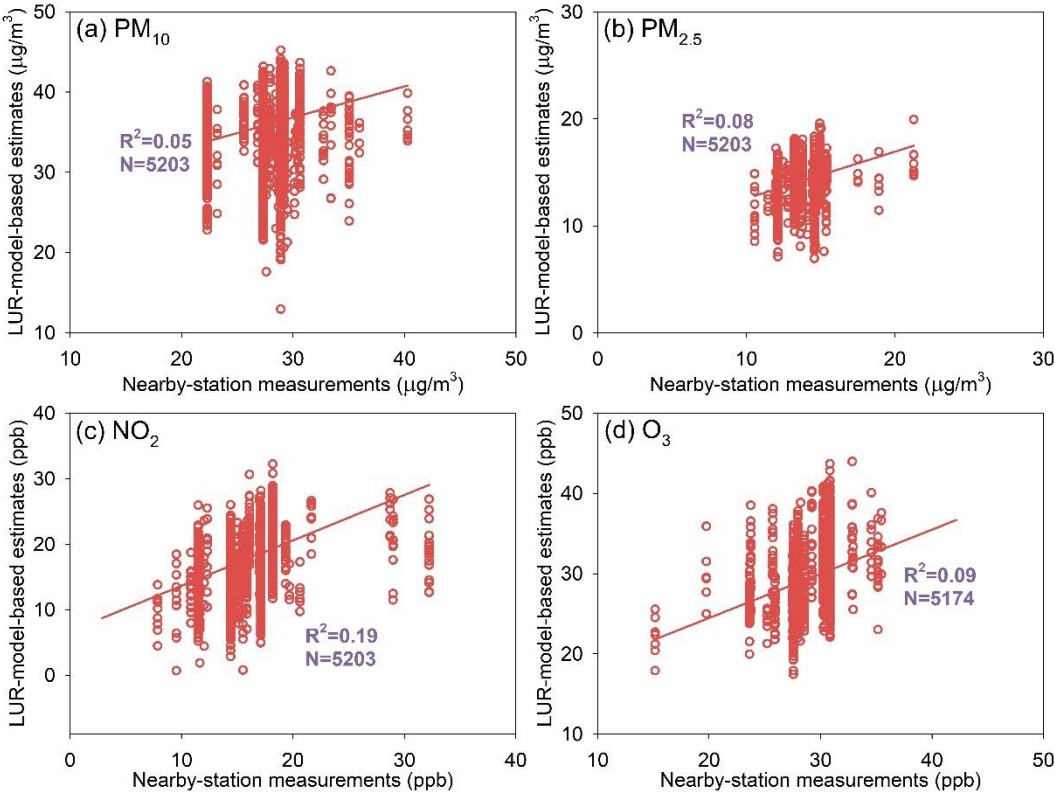

**Figure 6.** The linear regression of nearby-station air pollutant measurements and LUR-model-based air pollutant concentration estimates. (a) PM$_{10}$, (b) PM$_{2.5}$, (c) NO$_2$, and (d) O$_3$.