# Peer review of "Development and intercity transferability of land-use regression models for predicting ambient $PM_{10}$ , $PM_{2.5}$ , $NO_2$ and $O_3$ concentrations in northern Taiwan"

_Atmospheric Chemistry and Physics, 2020_

## Referee Comment (RC1) · Anonymous Referee #2 · 13 Nov 2020

General comment: This study addressed the important research question about air quality modeling for epidemiological studies and established a well-validated LUR model. The authors have provided reasonable responses to my comments in the first round of review. Here are my further comments. I recommend publishing this paper after addressing the following minor comments.

Specific comments: Line 16: "develop" -> "developed" Line 16: "evaluate" -> "evaluated" Line 23: this sentence should be revised for better readability. "with R2 and leave-one-out cross-validation (LOOCV) R2 values of > 0.72 and > 0.53, respectively."

-> "with R2 of > 0.72, and leave-one-out cross-validation (LOOCV) R2 values of > 0.53." Line 30: "our study" -> "this study" Line 41: I do not think LUR is a standard modeling approach. It is just a typical approach. So, it is suggested to revise the sentence to be "land-use regression (LUR) is a widely used modeling approach to characterize long-term average air pollutant concentrations" Line 46: "these stations" -> "the stations" Line 55: "have been" -> "were" Line 63: "is that the established models are usually only valid during the measurement period" -> "is that the established models only reflect the situation during the measurement period" Line 73: what does the "they" mean? Does it mean the previous two studies cited before the sentence? Line 79: "The remainder of this paper . . ." -> "This paper . . ." Line 119: "require" -> "requires" Line 119: "is" -> "was" Line 301: the opposite trend of NO2 and O3 is definitely the O3 titration in urban areas. This should be mentioned here. Table 3: the empty grids should be filled by grey color. Figure 2: Green color is not a good choice for display and should not be used. I suggest the authors to change the green dots to black or blue so as to enhance the readability. The size of the texts in the figure should be enhanced. Figure 4: the size of the texts is too small.

---

## Referee Comment (RC2) · Anonymous Referee #1 · 24 Dec 2020

Air pollution exposure estimates are highly dependent on the high-density observational network of atmospheric pollutants. Nevertheless, due to the limitation of factors such as harsh terrain, economic development level, and road accessibility, the routine air quality measurement sites are extremely lacking and calls for novel methods to fill out the data gap. The manuscript by Li et al. attempted to develop land-use regression (LUR) models to estimate several major atmospheric pollutants and to evaluate the transferability of LUR models between nearby cities in northern Taiwan region. To the best of my knowledge, this is one of a kind given its uniqueness in considering the

predictor variables and forward regression methods applied. Meanwhile, I noticed that transferability of the city-specific air pollutant LUR models exhibit large uncertainties except for NO2 LUR model, indicating there is still large room to be improved. Overall, this manuscript is logically structured and well written. Therefore, I recommend a minor revision before its acceptance for publication in ACP. The following is my specific comments that need to be addressed. Specific comments: 1. The transferability of LUR models across areas or cities varies greatly, given the large regional variability of predictive performances of LUR models. The authors are encouraged to strengthen the motivation why they developed one model to predict the air quality in northern Taiwan, or highlight/discuss the strengths of their LUR model, compared with previously established model. 2. L33-34: The health effect of aerosol is not adequately cited since air pollution has been well recognized to adversely affect cardiovascular diseases. The authors are suggested to consider citing Sun et al. 2011 (doi: 10.1161/CIRCULATIONAHA.109.893461); Yin et al. 2020 (doi: 10.1021/acs.estlett.9b00735) 3. L39: "estimating" -> "estimate" 4. L213: "Traffic emission is a major source of air pollution in urban areas of the TKMA (Lee et al., 2014; Wu et al., 2017)." Please be more specific regarding the contribution of traffic emission to air pollution in TKMA, e.g., what is the percentage? 5. L317-320: I am confused with the logic that the weak correlation between air pollution LUR model derived results and nearby-station measurements (Figure 6), makes the author believe in the notion that thereby air pollution LUR models may provide more accurate exposure estimates than nearby-station measurements. Please clarify it.

---

## Author Comment (AC1) · 29 Dec 2020

**#Referee 2**

Thanks the reviewer for the valuable comments. We have revised the manuscript accordingly, with a point-by-point reply to the comments, and a marked-up manuscript version showing the changes made (text in red).

- Line 16: "develop" -> "developed" Line 16: "evaluate" -> "evaluated"
  **Response:** Revised as suggested.
  P1, Line 17: "**In this study, we developed and evaluated…**"

- Line 23: this sentence should be revised for better readability. "with R2 and leave-one-out cross-validation (LOOCV) R2 values of > 0.72 and > 0.53, respectively." ->"withR2of>0.72,andleave-one-outcross-validation(LOOCV)R2valuesof>0.53."
  **Response:** Revised as suggested.
  P1, Line 24-25: "**with $R^2$ values of > 0.72 and leave-one-out cross-validation (LOOCV) $R^2$ values of > 0.53.**"

- Line 30: "our study" -> "this study"
  **Response:** Revised as suggested.
  P1, Line 31: "**…, this study is the first to…**"

- Line 41: I do not think LUR is a standard modeling approach. It is just a typical approach. So, it is suggested to revise the sentence to be "land-use regression (LUR) is a widely used modeling approach to characterize long-term average air pollutant concentrations"
  **Response:** Agreed and revised.
  P2, Line 42-43: "**…, land-use regression (LUR) is a widely used modeling approach…**"

- Line 46: "these stations" -> "the stations"
  **Response:** Done.
  P2, Line 48: "**… surrounding the stations…**"

- Line55: "have been"->"were"
  **Response:** Revised.
  P2, Line 57: "**… in the Taiwan region were limited…**"

- Line63: "is that the established models are usually only valid during the measurement period" -> "is that the established models only reflect the situation during the measurement period"
  **Response:** Done.
  P2, Line 65-66: "**… may only reflect the situation…**"

- Line 73: what does the "they" mean? Does it mean the previous two studies cited before the sentence?
  **Response:** The mentioned word was revised to make the meaning clearer.
  P3, Line 75: "**Previous studies on the transferability of LUR models…**"

- Line 79: "The remainder of this paper ..." -> "This paper ..."
  **Response:** We decide to keep the phrase "the remainder of this paper…" because the Introduction section is not included here.

- Line 119: "require" -> "requires"
  **Response:** It is correct to use "require" here because the subject of this sentence is "…estimates…".
  P4, Line 120-121: "**Daily and annual average estimates for the air pollutants require…**"

- Line 119: "is" -> "was"
  **Response:** Revised.
  P4, Line 121: "**...; otherwise there was no value…**"

- Line 301: the opposite trend of NO2 and O3 is definitely the O3 titration in urban areas. This should be mentioned here.
  **Response:** The related information was added.
  P10, Line 303-304: "**… were negatively correlated because of the strong NO$_x$ titration effect in urban areas…**"

- Table 3: the empty grids should be filled by grey color.
  **Response:** Modified as suggested.

- Figure 2: Green color is not a good choice for display and should not be used. I suggest the authors to change the green dots to black or blue so as to enhance the readability. The size of the texts in the figure should be enhanced.
Response: Thanks the reviewer for the comment. We changed the green color to blue and enlarged the text sizes.

[Figure]

Figure 2

- Figure 4: the size of the texts is too small.
  Response: We replotted Fig. 4 with the size of the texts enlarged.

[Figure]

Figure 4

---

## Author Comment (AC2) · 29 Dec 2020

**#Referee 1**

Thanks the reviewer for the valuable comments. We have revised the manuscript accordingly, with a point-by-point reply to the comments, and a marked-up manuscript version showing the changes made (text in red).

- 1. The transferability of LUR models across areas or cities varies greatly, given the large regional variability of predictive performances of LUR models. The authors are encouraged to strengthen the motivation why they developed one model to predict the air quality in northern Taiwan, or highlight/discuss the strengths of their LUR model, compared with previously established model.
  **Response:** In our original manuscript, we did include the motivation part (Line 57-68 and Line 15-17).

  P2, Line 57-68: "**In addition, most previous Taiwan LUR studies used data from purpose-designed monitoring networks or combined purpose-designed and routine monitoring networks (Ho et al., 2015; Lee et al., 2014; Lee et al., 2015). …As a result, a general limitation of LUR models upon purpose-designed monitoring networks is that the established models may only reflect the situation the measurement period (Hoek et al., 2008; Shi et al., 2020). Therefore, the development of long-term average LUR models for specific air pollutants using only routine monitoring networks should be explored, which is especially critical for epidemiological studies.**"
  P1, Line 15-17: "**To provide long-term air pollutant exposure estimates for epidemiological studies, it is essential to test the feasibility of developing land-use regression (LUR) models using only routine air quality measurement data and to evaluate the transferability of LUR models between nearby cities.**"

- 2. L33-34: The health effect of aerosol is not adequately cited since air pollution has been well recognized to adversely affect cardiovascular diseases. The authors are suggested to consider citing Sun et al. 2011 (doi: 10.1161/CIRCULATIONAHA.109.893461); Yin et al. 2020 (doi: 10.1021/acs.estlett.9b00735)
  **Response:** We added these two articles as references (Line 34-35, Line 507-508, and Line 553-555).

  P2, Line 34-35: "**…such as lung function, and respiratory and cardiovascular diseases (Çapraz et al., 2017; Sun et al., 2010; Yin et al., 2020; Zhou et al., 2020).**"

  P16, Line 507-508: **Sun, Q., Hong, X. and Wold, L.E.: Cardiovascular effects of ambient particulate air pollution exposure. Circulation 121(25), 2755-2765, 2010.**

  P17, Line 553-555: **Yin, P., Guo, J., Wang, L., Fan, W., Lu, F., Guo, M., Moreno, S.B., Wang, Y., Wang, H., Zhou, M. and Dong, Z.: Higher risk of cardiovascular disease associated with smaller size-fractioned particulate matter. Environ. Sci. Technol. Lett. 7(2), 95-101, 2020.**

- 3. L39: "estimating" -> "estimate"
  **Response:** Revised as suggested.

  P2, Line 41: "**… estimate population exposure…**"

- 4. L213: "Traffic emission is a major source of air pollution in urban areas of the TKMA (Lee et al., 2014; Wu et al., 2017)." Please be more specific regarding the contribution of traffic emission to air pollution in TKMA, e.g., what is the percentage?
  **Response:** We did include the contribution percentage of traffic emission to air pollution ($PM_{2.5}$ in the cited study) in TKMA (Line 216-218).

  P7, Line 216-218: "**For instance, it was reported that gasoline and diesel vehicle emissions contributed approximately half of $PM_{2.5}$ concentrations in Taipei City based on source apportionment analysis (Ho et al., 2018).**"

- 5. L317-320: I am confused with the logic that the weak correlation between air pollution LUR model derived results and nearby-station measurements (Figure 6), makes the author believe in the

notion that thereby air pollution LUR models may provide more accurate exposure estimates than nearby-station measurements. Please clarify it.

**Response:** We have corrected our text to make the meaning clearer (Line 323-328).

P11, Line 323-328: "**A possible explanation is that LUR-model-based exposure estimates generally accounted for neighbourhood-scale variations of air pollutant concentrations, while the nearby-station measurements usually only revealed the urban-scale variability of air pollution (e.g., urban area versus suburban area versus rural area) (Marshall et al., 2008). The LUR-model-based exposure estimates and nearby-station measurements should be further validated if the air quality measurement data at residential locations of cohort participants (if not all, at least some of the participants) are available.**"